# L-Theanine Prolongs the Lifespan by Activating Multiple Molecular Pathways in Ultraviolet C-Exposed *Caenorhabditis elegans*

**DOI:** 10.3390/molecules29112691

**Published:** 2024-06-06

**Authors:** Liangwen Chen, Guijie Chen, Tingting Gai, Xiuhong Zhou, Jinchi Zhu, Ruiyi Wang, Xuemei Wang, Yujie Guo, Yun Wang, Zhongwen Xie

**Affiliations:** 1State Key Laboratory of Tea Plant Biology and Utilization, School of Tea and Food Sciences and Technology, Anhui Agricultural University, Hefei 230036, China; chenliangwen123@163.com (L.C.);; 2Key Laboratory of Bioresource and Environmental Biotechnology of Anhui Higher Education Institutes, School of Biological Engineering, Institute of Digital Ecology and Health, Huainan Normal University, Huainan 232001, Chinajinchizhu@foxmail.com (J.Z.);

**Keywords:** L-theanine, UVC, mitochondrial DNA damage, *C. elegans*, lifespan

## Abstract

L-theanine, a unique non-protein amino acid, is an important bioactive component of green tea. Previous studies have shown that L-theanine has many potent health benefits, such as anti-anxiety effects, regulation of the immune response, relaxing neural tension, and reducing oxidative damage. However, little is known concerning whether L-theanine can improve the clearance of mitochondrial DNA (mtDNA) damage in organisms. Here, we reported that L-theanine treatment increased ATP production and improved mitochondrial morphology to extend the lifespan of UVC-exposed nematodes. Mechanistic investigations showed that L-theanine treatment enhanced the removal of mtDNA damage and extended lifespan by activating autophagy, mitophagy, mitochondrial dynamics, and mitochondrial unfolded protein response (UPR^mt^) in UVC-exposed nematodes. In addition, L-theanine treatment also upregulated the expression of genes related to mitochondrial energy metabolism in UVC-exposed nematodes. Our study provides a theoretical basis for the possibility that tea drinking may prevent mitochondrial-related diseases.

## 1. Introduction

Energy animates life. Mitochondria are multi-functional organelles within eukaryotic cells and generate the most energy by producing adenosine 5′-triphosphate (ATP) via oxidative phosphorylation (OxPHOs), which functions in fatty acid oxidation, apoptosis, the cell cycle, and cell signaling [1]. The majority of enzymes necessary for oxidative phosphorylation (OxPHOs) are encoded by nuclear DNA (nDNA), while a specific subset of 13 essential protein subunits of respiratory complexes I, III, IV, and V, along with two rRNAs and 22 tRNAs, are exclusively encoded by mitochondrial DNA (mtDNA) [2]. mtDNA is more susceptible to damage compared to nDNA due to its lack of nucleotide excision repair (NER), which typically repairs helix-distorting damage caused by common environmental factors such as polycyclic aromatic hydrocarbons (PAHs), mycotoxins, and ultraviolet C radiation (UVC) [3]. Previous studies reported that mtDNA is dozens of times more sensitive to PAHs than nDNA [4], and helix-distorting mtDNA lesions are persistent and cause a decrease in mtDNA replication and transcription [5]. Natarelli et al. have shown that persistent mtDNA damage can disrupt mitochondrial function [6]. A growing body of research indicates that mitochondrial dysfunction and mutations in mtDNA are linked to a range of human pathologies, including cancer, type 2 diabetes mellitus, neurodegenerative disorders, mitochondrial diseases, and aging [7,8,9,10,11]. Therefore, the integrity of mtDNA is very important to the entire organism.

The nematode *Caenorhabditis elegans* (*C. elegans*), commonly found in decaying organic material such as leaf litter, gained prominence in the scientific community due to Sydney Brenner’s groundbreaking research on its development and neurobiology in 1965 [12]. This organism is widely utilized as a primary model for studying molecular-level processes related to aging, stress resistance, mitochondrial biology, and apoptosis [13,14]. This is primarily attributed to its short lifespan of approximately 15–21 days, its comprehensive sequencing and annotation of all genes [15], and the availability of numerous mutant strains [16]. Importantly, findings from studies conducted on *C. elegans* have been shown to have predictive value for outcomes in higher organisms [17]. Furthermore, there is minimal disparity in mitochondrial biology between *C. elegans* and humans [14]. The mitochondrial genome of *C. elegans* is notably smaller than that of humans, with 13.8 Kb base pairs compared to 16.6 Kb [18]. While the majority of encoded genes are similar between the two species, the *atp-8* gene remains unidentified in *C. elegans* [19]. To investigate the enduring consequences of helix-distorting mtDNA damage, Bess AS et al. employed UVC radiation to induce such damage in *C. elegans*, creating a system in which only mtDNA damage was observed as nuclear DNA damage was repaired by nucleotide excision repair (NER). The researchers also observed that sustained mtDNA damage led to L3 larval arrest in *C. elegans*, with the severity of mtDNA damage positively associated with the frequency of arrest. Consequently, *C. elegans* serves as a valuable model for examining the effects of persistent mtDNA damage in vivo.

Previous research has demonstrated a strong association between psychological stress and various health outcomes, including mood swings, depression, cardiovascular disorders, and different types of cancer [20]. To cope with these negative effects of psychological stress, individuals are increasingly opting for natural health supplements over synthetic products. Green tea is a traditional drink globally due to its favorable taste and relaxing effects on the body [21]. The consumption of green tea has been linked to numerous health benefits, leading to increased research interest in the health properties of specific components of green tea [22]. L-theanine, also referred to as L-γ-glutamylethylamide, is a non-protein amino acid found in green tea that serves as a significant bioactive compound [23]. It accounts for approximately 50% of the total free amino acids in green tea and represents 1–2% of the tea’s dry weight [24]. Additionally, L-theanine has been deemed safe and non-toxic as a food additive by the United States Food and Drug Administration (US FDA) [25]. Previous studies have demonstrated the various health benefits associated with L-theanine, such as its neuroprotective and anxiolytic effects, its ability to regulate immune response, and its capacity to relax neural tension [26]. Consequently, L-theanine is frequently incorporated as a key component in functional beverages and dietary supplements. Pre-administration of L-theanine to mice has been shown to notably diminish irinotecan-induced genetic harm in bone marrow cells [27]. Nonetheless, there is still a lack of comprehensive knowledge on the potential of L-theanine to improve the elimination of mtDNA damage in organisms. Consequently, this study aimed to examine the impact of L-theanine treatment on UVC-induced mtDNA damage and its underlying molecular mechanisms in *C. elegans*.

## 2. Results

### 2.1. L-Theanine Enhanced the Removal of UVC-Induced mtDNA Damage

Bess et al. reported that UVC-induced mtDNA damage leads to L3 larval arrest, and the proportion of L3 larval arrest was positively correlated with mtDNA damage in *C. elegans* [3]. Therefore, the L3 arrest phenotype serves as an important indicator of mtDNA damage. Our results showed that a UVC dose-dependent increase in L3 arrest was observed in UVC-exposed L1 nematodes grown on plates with food for 48 h at 7.5 J/m^2^ start dosage (Figure 1A). When the UVC exposure dose was set up at 12.5 J/m^2^, mtDNA damage was detected, while nDNA damage could not be detected in nematodes (Figure 1B). Therefore, we used 12.5 J/m^2^ as a dose that only induced mtDNA damage with UVC exposure in the experiment. Previous research has demonstrated that the pretreatment of mice with L-theanine resulted in a significant reduction of irinotecan-induced genomic damage in bone marrow cells [27]. This suggests that L-theanine may possess chemoprotective properties against DNA damage in organisms. However, there is limited understanding regarding the potential of L-theanine to enhance the removal of UVC-induced mtDNA damage in organisms. In order to investigate the impact of L-theanine on the removal of UVC-induced mtDNA damage, UVC-exposed (12.5 J/m^2^) L1 nematodes were treated with varying concentrations of L-theanine for 48 h, and L3 arrest was evaluated using a dissection microscope. The findings indicate that treatment with L-theanine within the range of 10–400 µg/mL significantly decreased L3 arrest, with 50 µg/mL of L-theanine demonstrating the most pronounced effect (Figure 1C). At the same time, the mtDNA damage was obviously reduced by treatment with 50 µg/mL L-theanine (Figure 1D). Therefore, 50 µg/mL of L-theanine was applied in the following experiments unless stated otherwise. The above results suggested that L-theanine treatment may have a role in removing UVC-induced mtDNA damage in *C. elegans*. In order to rule out the effect of L-theanine on L3 arrest and the development of nematodes, L1 nematodes were placed in Petri dishes with various concentrations of L-theanine for 48 h, and then L3 arrest and body length of nematodes were determined. L-theanine treatment in the range of 10–400 µg/mL had no effect on L3 arrest and body length of nematodes (Figure 1E,F). The L1 nematodes were exposed to different doses of UVC and were then transferred to 50 µg/mL L-theanine treatment for 48 h to count the ratio of L3 arrest. Appendix A showed that L3 arrest of L1 nematodes with 50 µg/mL L-theanine treatment was significantly reduced at 10 J/m^2^ and 12.5 J/m^2^ UVC, in which the more significant reduction was observed at 12.5 J/m^2^ UVC. L3 arrest of L1 nematodes did not significantly change at 15 J/m^2^ and 20 J/m^2^ UVC. Therefore, the combination of UVC (12.5 J/m^2^) with L-theanine (50 µg/mL) was used in all subsequent experiments unless stated otherwise. In order to investigate the potential role of L-theanine in enhancing nucleotide excision repair (NER) in nDNA to mitigate UVC-induced L3 arrest in *C. elegans*, a series of mutant worms with impaired NER pathways were utilized as UVC-exposed L1 nematodes. The results depicted in Appendix A indicate a significant reduction in L3 arrest of L1 nematodes treated with L-theanine in the NER pathway mutants (*xpa-1*, *polh-1*, and *xpf-1*). These findings suggest that L-theanine treatment did not impact NER in nDNA to alleviate UVC-induced L3 arrest in *C. elegans*. Taken together, these findings indicate that treatment with L-theanine improves the removal of UVC-induced mtDNA damage in *C. elegans*.

### 2.2. L-Theanine Extended the Lifespan and Enhanced the Heat-Stress Resistance of UVC-Exposed Nematodes

Mitochondria are recognized as pivotal contributors to the aging process, with age-related decline in mitochondrial function being a common occurrence [11]. The preservation of mtDNA integrity is essential for maintaining optimal mitochondrial function, as sustained damage to mtDNA can lead to mitochondrial dysfunction [11]. Studies have demonstrated a shortened lifespan in *C. elegans* with impaired mitochondria. To assess the impact of L-theanine on the longevity of UVC-exposed nematodes, UVC-exposed L1 nematodes were treated with or without L-theanine, followed by lifespan measurements. Compared to the UVC-exposed group, the L-theanine-treated group significantly increased longevity (Figure 2A). This suggests that L-theanine treatment enhanced the lifespan in UVC-exposed *C. elegans*.

Persistent damage to mtDNA results in the halting of both replication and transcription processes, ultimately leading to a reduction in mtDNA levels and proteins, as well as the onset of mitochondrial dysfunction [28]. Palikaras et al. observed a decrease in heat-stress resistance in *C. elegans* with mitochondrial dysfunction [29]. To evaluate the effects of L-theanine treatment on the heat-stress resistance of UVC-exposed nematodes, experiments involving heat-stress conditions were carried out. The heat-stress resistance assay involved treating UVC-exposed L1 nematodes with or without L-theanine for either 24 or 48 h, followed by exposure to heat shock at 35 °C for 7 h. The findings indicated a significant improvement in the heat-stress resistance of UVC-exposed nematodes following treatment with L-theanine at both 24 and 48 h (Figure 2B,C). This suggests L-theanine treatment enhanced heat-stress resistance in UVC-exposed *C. elegans*.

### 2.3. L-Theanine Improved Mitochondrial Morphology of UVC-Exposed Nematodes

Mitochondria exhibit dynamic tubular structures that undergo continuous reshaping through processes such as biogenesis, fission, fusion, mitophagy, and motility in a regulated manner, ultimately determining the morphology of the mitochondrial network. Alterations in mitochondrial morphology have been implicated in cellular responses to stress and pathological conditions in nematodes [30]. Given the crucial role of mitochondrial dynamics in the removal of mtDNA damage, our study examined the effects of L-theanine on the mitochondrial morphology of UVC-exposed L1 nematodes. To assess the mitochondrial morphology, L1 nematodes exposed to UVC radiation and harboring an extrachromosomal Pmyo-3::matrixGFP and expressing mito-GFP in muscle cells were transferred onto dish plates supplemented with 50 µg/mL L-theanine. After 2, 4, and 6 days, pictures were taken of the treated nematodes with confocal fluorescence microscopy. Our observations revealed a time-dependent increase in mitochondrial fragmentation and a decrease in the tubular mitochondria proportion in UVC-exposed nematodes (Figure 3A–D). However, our findings demonstrate that treatment with L-theanine effectively mitigated the decline in the tubular mitochondria portion in UVC-exposed *C. elegans* starting from the fourth day compared to UVC-treated worms (Figure 3A–D). These results suggest that L-theanine treatment significantly enhances mitochondrial morphology in UVC-exposed *C. elegans*.

### 2.4. L-Theanine Elevated Steady-State ATP Level in UVC-Exposed Nematodes

One of the key roles of mitochondria is the production of energy via oxidative phosphorylation (OxPHOS). The essential protein subunits of respiratory complexes I, III, IV, and V in the OxPHOS system are encoded by mtDNA. Thus, mtDNA damage will reduce steady-state ATP levels in cells [3]. Research by Leung MC et al. has shown that the steady-state ATP level of UVC-exposed L1 nematodes was reduced at later larval development in *C. elegans* [31]. To assess the potential of L-theanine treatment in enhancing ATP levels in UVC-exposed L1 larvae during later developmental stages, ATP levels were quantified in vivo using a validated transgenic strain (PE255) expressing a luciferase gene. The results showed that the L-theanine-treated group exhibited higher ATP levels at the 24 h and 48 h timepoints compared to the UVC-exposed group (Figure 4A–H). This indicates that L-theanine treatment increased the steady-state ATP level of UVC-exposed L1 at later larval development.

### 2.5. Autophagy, Mitochondrial Dynamics, and UPR^mt^ Mediated the Reduction of mtDNA Damage and Extension of Lifespan in UVC-Exposed C. elegans Treated with L-Theanine

The preservation of mtDNA integrity is crucial for maintaining proper mitochondrial function and is highly susceptible to harm caused by various environmental factors such as polycyclic aromatic hydrocarbons (PAHs), mycotoxins, and ultraviolet C radiation (UVC) [3,25]. A previous investigation demonstrated that processes such as mitochondrial dynamics, mitophagy, and autophagy collectively participate in the elimination of persistent mtDNA damage in UVC-exposed *C. elegans* [3]. Therefore, we suspected that the removal of mtDNA damage by L-theanine treatment is due to the contribution of mitochondrial dynamics, mitophagy, and autophagy. To verify this possibility, L3 arrest was determined for several UVC-exposed mutant worms with deficiencies in autophagy (*unc-51*), mitophagy (*pink-1* and *pdr-1*), and mitochondrial dynamics (*fzo-1*, *eat-3*, and *drp-1*) following treatment with L-theanine. The results showed that L3 arrest was significantly inhibited by L-theanine treatment (Figure 5A,B). This suggested that the reduction of mtDNA damage by L-theanine may occur through mitochondrial dynamics, mitophagy, and autophagy. Research has demonstrated that the unfolded protein response is a critical stress response pathway that serves to protect the mitochondria. Furthermore, recent studies have indicated that the mitochondrial unfolded protein response (UPR^mt^) significantly contributes to the mitigation of mtDNA-related disorders in *C. elegans* [32]. So, does UPR^mt^ contribute to the removal of mtDNA damage by L-theanine treatment in UVC-exposed *C. elegans*? To test this hypothesis, we examined the effect of L-theanine treatment on the L3 arrest of several UVC-exposed mutant worms defective for UPR^mt^ (*atfs-1, haf-1*, and *ubl-5*) and found that the effect of L-theanine treatment on L3 arrest was inhibited (Figure 5C), indicating that UPR^mt^ plays a key role in the reduction of mtDNA damage by L-theanine treatment. Previous research has shown that L-theanine treatment prolongs the lifespan of wild-type nematodes exposed to UVC (Figure 2A). The life span of nematodes is closely related to the degree of damage to mtDNA [11]. In order to investigate the impact of mitophagy, mitochondrial dynamics, and UPR^mt^ on the longevity of nematodes subjected to UVC radiation and treated with L-theanine, we conducted an analysis of the lifespan of various mutant worms deficient in mitophagy (*pdr-1*), mitochondrial dynamics (*drp-1*), and UPR^mt^ (*atfs-1* and *haf-1*). Our findings indicate that the potential lifespan-extending effects of L-theanine treatment were impeded in these mutant worms (Figure 5D–G). The above results indicate that L-theanine treatment prolonged the lifespan of nematodes exposed to UVC, most likely by activating multiple pathways, including autophagy, mitochondrial autophagy, mitochondrial dynamics, and UPR^mt^, to enhance the clearance ability of mtDNA damage.

The previous results showed that the reduction of mtDNA damage by L-theanine treatment depended on autophagy, mitophagy, mitochondrial dynamics, and UPR^mt^. Therefore, the following study focused on the effect of L-theanine treatment on the gene expression of autophagy, mitophagy, and UPR^mt^ in UVC-exposed *C. elegans*. The results show that compared to the UVC treatment group, the L-theanine treatment group significantly upregulated the gene expression of *bec-1* (autophagy), *dct-1* (mitophagy), and *hsp-60* (UPR^mt^) without causing any changes to the gene expression of *lgg-1* (autophagy) and *atg-18* (autophagy) (Figure 6A). Additionally, subsequent agarose gel electrophoresis results demonstrated a notable enhancement in the expression levels of *bec-1*, *dct-1*, and *hsp-60* genes in the L-theanine treatment group (Figure 6B–D). These results further confirm that autophagy, mitophagy, and UPR^mt^ play a key role in reducing mtDNA damage and prolonging the lifespan of nematodes exposed to UVC by L-theanine treatment. Persistent mtDNA damage causes mitochondrial dysfunction, which ultimately impairs energy metabolism [33]. Hence, the study examined the expression of genes related to mitochondrial energy metabolism in UVC-exposed nematodes with and without L-theanine treatment. The results depicted in Appendix A demonstrate that the L-theanine treatment group exhibited a significant upregulation in the gene expression of *pcy-1* and *cox-4* compared to the UVC treatment group, whereas the gene expression of *cts-1* and *hxk-1* remained unchanged. These findings suggest that L-theanine treatment ameliorates the energy metabolism impairment resulting from mtDNA damage.

## 3. Discussion

Numerous research studies have highlighted the importance of maintaining mtDNA integrity in the context of human aging and disease [11,29,32]. The lack of nucleotide excision repair (NER) mechanisms, which are responsible for repairing DNA damage caused by environmental factors, can lead to the accumulation of persistent damage in mtDNA [3,25]. Bess AS et al. utilized a *C. elegans* model to explore the impact of UVC-induced mtDNA damage, uncovering a direct relationship between the level of mtDNA damage and the incidence of L3 larval arrest [3]. Therefore, L3 larval arrest is an important phenotypic indicator of mtDNA damage. Consistent with the research of Bess et al., our results showed that the L3 larval arrest was proportional to the dose of UVC in *C. elegans* (Figure 1A). Mitochondria are highly dynamic tubular organelles that are constantly remodeled by fusion and division, and their morphology is closely related to mtDNA rescue [3,34]. Research conducted by Bess AS et al. demonstrated that young adult nematodes that were subjected to UVC irradiation at a single dose of 50 J/m^2^ had no significant variability in mitochondrial morphology, observed at 24 h and 48 h post-exposure [3]. Interestingly, it was observed that the mitochondrial morphology of UVC-exposed L1 nematodes became fragmented and disorganized on the fourth day following exposure (Figure 3A). We speculate that the above results may be mainly due to different periods of UVC exposure for the nematodes, as well as the extended period of observation in comparison to prior research. Research conducted by Shokolenko et al. demonstrated that, while there was a rapid loss of mtDNA in Hela/2641 cells transduced with either the exoIII or mUNG1 construct, alterations in mitochondrial morphology were not evident within the initial four days but rather manifested after a period of 120 h [35]. These findings indicate that a temporal delay exists between mtDNA damage and the onset of morphological changes in mitochondria.

The integrity of mtDNA is crucial to the proper function of the mitochondrial respiratory chain. Most of the ATP in organisms is produced by oxidative phosphorylation conducted by the mitochondrial respiratory chain in mitochondria. Persistent mtDNA damage will lead to mitochondrial dysfunction, which further leads to a decreased capacity to produce ATP [35]. The study conducted by Leung MC et al. demonstrated a significant decrease in ATP production in UVC-exposed L1 nematodes compared to those without UVC treatment [18]. Our findings further support this conclusion, showing a significant reduction in ATP production in the UVC-exposed group compared to the unexposed group at both the 24 h and 48 h timepoints (Figure 4A–H). Thus, we concluded that L1 nematodes exposed to UVC are an excellent animal model for studying mtDNA damage, which can cause changes in mitochondrial morphology and reduce ATP content.

Mitochondria are the main sites for oxidative phosphorylation, supplying more than 90% of cellular ATP [36]. The few small subunits required for oxidative phosphorylation can only be encoded by mtDNA [37]. Reactive oxygen species (ROS) are generated through a variety of cellular processes, external influences, and/or diverse pathological conditions [38]. ROS have the potential to induce oxidative injury to biological macromolecules such as nucleic acids, proteins, and lipids, with mtDNA being particularly susceptible [38]. Damaged mtDNA increases the production of oxygen free radicals and further exacerbates mitochondrial dysfunction. Therefore, some researchers pay attention to some natural active substances that can protect mtDNA and have an antioxidant effect. A variety of antioxidant substances have been found to improve mtDNA damage caused by oxidative stress, such as curcumin [39], resveratrol [40], vitamins C and E [41], *Phellinus linteus* [42], and catechins [43]. Oxidative stress-induced damage to mtDNA can be mitigated through the base excision repair pathway, which is effective in removing oxidized bases [44]. However, the repair of helix-distorting damage to mtDNA caused by common environmental factors is challenging due to the absence of nucleotide excision repair mechanisms. To date, there are few reports that natural active substances are helpful in removing helix-distorting mtDNA damage. L-theanine (L-γ-glutamylethylamide) is a non-protein amino acid found in green tea and is recognized for its bioactive properties [26]. Numerous research studies have shown that L-theanine has numerous potent health benefits, including neuroprotective effects, anxiolytic effects, regulating the immune response, and relaxing neural tension [26]. To date, there is a lack of research regarding the impact of L-theanine on mtDNA damage. In the current investigation, it was observed that L-theanine administration notably decreased UVC-induced L3 larval arrest, suggesting its potential to mitigate UVC-induced mtDNA damage in *C. elegans* (Figure 1B). Additionally, the study demonstrated that L-theanine treatment improved mitochondrial morphology and elevated ATP levels in UVC-exposed *C. elegans*. Therefore, we concluded that L-theanine treatment enhances the ability to remove mtDNA damage and improve mitochondrial function in UVC-exposed *C. elegans*.

Autophagy and mitochondrial dynamics, including fission, fusion, and mitophagy, are crucial mechanisms for eliminating damaged mitochondria, enhancing overall mitochondrial function, and maintaining mitochondrial continuity [45]. The study conducted by Bess et al. demonstrated that mitochondrial dynamics and autophagy are involved in the clearance of UVC-induced mtDNA damage in *C. elegans*, with prolonged mtDNA damage leading to compromised mitochondrial function [3]. Mitochondrial fusion and autophagy act as protective mechanisms against dysfunction by promoting functional complementation, mtDNA replication, and the removal of dysfunctional mitochondria. Conversely, mitochondrial fission and autophagy contribute to the elimination of mtDNA damage, with mitochondrial fusion indirectly supporting this process through the maintenance of crucial mitochondrial function and mtDNA replication [3]. Mitophagy is induced in response to stress to remove dysfunctional mitochondria and support cell viability. Simultaneous activation of SKN-1 augments mitophagy and facilitates mitochondrial biogenesis by upregulating the expression of DCT-1 and various mitochondrial genes [29]. FZO1 is a necessary protein to control outer mitochondrial membrane fusion events. EAT-3 serves as the ortholog of OPA1, playing a crucial role in inner mitochondrial membrane fusion and maintenance of cristae structure. DRP1 is a key protein involved in mitochondrial fission. UNC51 functions as a serine/threonine kinase in autophagy induction, while BEC1, a class III phosphatidylinositol 3-kinase (PI3K), aids in the recruitment of other autophagy proteins to pre-autophagosomal structures. PINK1, a serine/threonine-protein kinase, is a critical and conserved component of mitophagy. DCT-1, a protein hypothesized to interact with mammalian NIX/BNIP3L, bnip 3 (nip 3-like protein), and adenovirus E1B, functions as a mitophagy receptor in mammals. *Pdr-1* is tasked with encoding the *Caenorhabditis elegans* homolog of the parkin ubiquitin ligase PARK2, and the *pdr-1* (gk448) allele contains a partial deletion that notably reduces mitophagy in the nematode [32]. Studies have demonstrated that mutations in these genes lead to impaired autophagy, fission, and fusion in *C. elegans* [46]. The current investigation revealed that mutations in genes associated with autophagy (*unc-51*), mitophagy (*pink-1, pdr-1*), fusion (*fzo-1, eat-3*), and fission (*drp-1*) all led to a reduced efficacy of L-theanine treatment in promoting the removal of UVC-induced mtDNA damage (Figure 5A,B). These findings suggest that autophagy and mitochondrial dynamics play a crucial role in mediating the reduction of mtDNA damage with L-theanine treatment. Additionally, the findings from quantitative real-time PCR analysis indicate that treatment with L-theanine led to a significant increase in the mRNA expression of autophagy genes (*bec-1, lgg-1*) and the mitophagy gene (*dct-1*) in UVC-exposed *C. elegans* (Figure 6). This aligns with a recent study that demonstrated that resveratrol could mitigate rotenone-induced mtDNA damage by modulating mitochondrial dynamics [40]. Thus, our study suggests that L-theanine treatment can ameliorate mitochondrial dysfunction resulting from mtDNA damage by enhancing autophagy and mitochondrial dynamics to facilitate the removal of mtDNA damage in UVC-exposed *C. elegans*. 

It has been documented that stresses inducing mitophagy can concurrently trigger the activation of the mitochondrial unfolded protein response (UPR^mt^) [47]. The UPR^mt^ serves as a signaling pathway from the mitochondria to the nucleus, initiated by the buildup of unfolded proteins within the mitochondria, and ultimately leading to the upregulation of mitochondrial protective genes such as molecular chaperones and proteases [48,49]. Prior studies have demonstrated that the activation of UPR^mt^ can ameliorate neuromuscular impairments in *polg-1*(srh1) worms afflicted with mtDNA depletion, suggesting the potential therapeutic utility of UPR^mt^ in addressing mitochondrial dysfunction [32]. Various factors have been identified as necessary for the induction of UPR^mt^, including the HAF-1 peptide exporter, the CLPP-1 protease, a ubiquitin-like protein UBL-5, and two transcription factors, DVE-1 and ATFS-1 (ZC376.7) [50]. Consequently, mutants associated with UPR^mt^ (*atfs-1*, *haf-1*, and *ubl-5*) were utilized in this investigation, revealing that the ability of L-theanine treatment to enhance the removal of UVC-induced mtDNA damage was impeded (Figure 5C). Moreover, the mRNA expression of the UPR^mt^ gene (*hsp-60*) was significantly upregulated in the L-theanine treatment group (Figure 6). Hence, we concluded that L-theanine treatment enhances UPR^mt^ to improve mitochondrial dysfunction induced by mtDNA damage in UVC-exposed *C. elegans*.

Aging is characterized by the progressive accumulation of damaged lipids and proteins, resulting in cellular dysfunction, tissue and organ failure, and, ultimately, death [11]. Impaired mitochondrial maintenance is a common feature in both human pathologies and aging, affecting various cell types [51]. A study demonstrated that the accumulation of mitochondrial DNA damage correlates with the degree of tissue aging in mammals [11]. Our research revealed that treatment with L-theanine not only enhances the survival of UVC-exposed nematodes under heat stress but also prolongs the lifespan of UVC-exposed nematodes under normal culture conditions (Figure 2). Therefore, we hypothesized that the extended lifespan of UVC-exposed nematodes was mainly due to L-theanine’s ability to remove UVC-induced mtDNA damage and improve mitochondrial dysfunction induced by mtDNA damage in UVC-exposed *C. elegans*.

Generally, here we canonically demonstrated that L-theanine treatment extended the lifespan of UVC-exposed nematodes mainly by enhancing the ability to eliminate mtDNA damage to increase ATP production and improve mitochondrial morphology in UVC-exposed nematodes. The primary mechanism of L-theanine action is preliminarily clarified, as shown in Figure 7.

## 4. Materials and Methods

### 4.1. Reagents

L-theanine with a purity of 98% (Sigma, St. Louis, MO, USA) was dissolved in distilled water, sterilized through filtration using 0.2 μm pore size membranes, and subsequently stored at −20 °C for future utilization. All additional chemicals and reagents were procured from Sigma-Aldrich unless specified otherwise.

### 4.2. C. elegans Strains and Culture Conditions

*C. elegans* strains were maintained at 20 °C on K agar plates seeded with *OP50* strain *Escherichia coli* unless stated otherwise [52]. However, JK1107 (*glp-1(q224)* III was maintained at 15 °C. All *C. elegans* strains were obtained from the Caenorhabditis Genetics Center (University of Minnesota, MN, USA). They are N2 (Bristol, wild type), CB369: *unc-51(e369)*V, RB2547: *pink-1(ok3538)*II, VC1024: *pdr-1(gk448)* III, CU5991: *fzo-1(tm1133)* II, DA631: *eat-3(ad426)* II, CU6372: *drp-1(tm1108)* IV, QC115: *atfs-1(et15)* V, VC2654: *ubl-5(ok3389)* I, RB867: *haf-1(ok705)* IV, PE255: *feIs5* [sur-5p::luciferase::GFP + rol-6(su1006)] X, and SJ4103: *zcIs14* [myo-3::GFP(mit)].

In all experimental procedures, synchronized L1 larvae were obtained using a bleach-sodium hydroxide isolation method for eggs [53]. The specified concentration of L-theanine was achieved by applying a coating of L-theanine diluted in live *E. coli* OP50 suspension to the surface of the dry K agar medium [54].

### 4.3. UVC Exposure

UVC exposures were carried out within a specially designed exposure cabinet. The dose intensity of radiation was quantified using a UVX digital radiometer (UVP, Upland, CA, USA) to determine the necessary exposure time to achieve the desired dose. Synchronized L1 worms were placed on bacteria-free plates and exposed to the specified dose. Subsequently, under sterile conditions, the UVC-exposed L1 larvae were evenly distributed onto culture plates containing food at a minimum density of 300 worms per plate.

### 4.4. Lifespan Experiments and Heat-Stress Resistance Assays

Lifespan assays were conducted at a temperature of 20 °C using wild-type L1 larvae exposed to UVC. The larvae were placed on culture plates containing L-theanine at a minimum density of 40 worms per plate. In order to prevent the effects of offspring, the worms were transferred to new plates daily until all nematodes had perished. The viability of the worms was assessed by gently touching their pharynx with eyebrow-made picking needles.

Heat-stress resistance assays were performed at 35 °C with wild-type [55]. The L1 nematodes treated with UVC were placed on culture plates (Φ = 6 cm) containing L-theanine of various concentrations at 20 °C for 24 h and 48 h, then exposed to heat shock at 35 °C for 6 h and finally transferred to the incubator at 20 °C for 10 h before the number of nematode deaths was recorded. Under each concentration of L-theanine, 4 independent plates at a density of around 45 worms per plate were used and then repeated 3 times. The standard for nematode death is no response to a gentle touch.

### 4.5. L3 Arrest Analysis

The L1 nematodes, which had been exposed to UVC radiation, were distributed onto culture plates with a diameter of 6 cm, each containing varying concentrations of L-theanine at a density of approximately 400 worms per plate. Following a 48 h incubation period at 20 °C, the UVC-treated L1 nematodes were individually staged, and the percentage of worms progressing to the L4 stage was assessed based on the presence of the vulval crescent [3]. Given the noticeable difference in body length between L1, L2, and L3 larvae compared to L4 larvae, any larvae that did not progress to the L4 stage were classified as being in a state of L3 arrest. Unstressed wild-type nematodes will be in the young adult stage for approximately 2 h after 48 h of growth at 20 °C, while some mutants will suffer from delayed growth due to the lack of some genes. In order to eliminate the influence of delaying development on the proportion of L3 arrest, the unstressed mutants enter the young adult stage for around 2 h as a reference for the counting time.

### 4.6. mtDNA and nDNA Damage Measurements

All the specific steps of the experiments are followed according to the Gonzalez-Hunt CP et al. protocol [56]. The quantification of DNA damage is achieved by assessing the number of polymerase-inhibiting lesions through long amplicon quantitative PCR (LA-QPCR). Lesion frequency is determined by comparing amplification efficiency to undamaged controls. The DNA template for LA-QPCR was extracted directly from the lysate of six nematodes (N2). Primers used for LA-QPCR were as follows: mtDNA primers, forward-5′-CCA TCA ATT GCC CAA AGG GGA GT-3′, reverse-5′-TGT CCT CAA GGC TAC CAC CTT CTT CA-3′ (10.9 kb); nDNA primers, forward-5′-TGG CTG GAA CGA ACC GAA CCA T-3′, reverse-5′-GGC GGT TGT GGA GTG TGG GAA G-3′ (9.3 kb) [56]. In order to determine the copy number of mitochondria and the nuclear genome, we utilized worm glp-1 lysate as a substitute for the plasmid with a known copy number to establish a standard curve. This approach was chosen due to the inability of glp-1 worms to develop a germline at a temperature of 25 °C. The concentration of 40 μL of lysate from young adult glp-1 worms (equivalent to 20 worms) was found to be 1567 copies/µL. The primers employed for real-time PCR analysis were as follows: the mtDNA primers were forward-5′-AGC GTC ATT TAT TGG GAA GAA GAC-3′, reverse-5′-AAG CTT GTG CTA ATC CCA TAA ATG T-3′ (75 bp); and nDNA primers were forward-5′-GCC GAC TGG AAG AAC TTG TC-3′, reverse-5′-GCG GAG A TC ACC TTC CAG TA-3′ (164 bp) [56]. The quantification of amplified long PCR product serves as an indicator of lesion frequency, whereas the quantification of short PCR product serves as a means of normalizing to DNA concentration and genome copy number.

### 4.7. Body Length

The L1 nematodes were cultured on K agar plates (Φ = 6 cm) containing varying concentrations of L-theanine at 20 °C. After 48 h, 50 nematodes were randomly chosen from each experimental group and immobilized with 200 µL of 10 mM sodium azide solution for 5 min. Subsequently, the immobilized nematodes were imaged using a stereomicroscope equipped with a camera, and their lengths were quantified using Image J software (1.53i). The body length was determined by measuring the distance from the tip to the tail of each nematode [57].

### 4.8. Steady-State ATP Level Analysis

The PE255 nematode strain expressing luciferase was utilized for the investigation of ATP levels in live worms under relatively stable conditions [3]. Luminescence was measured using a 96-well microplate reader (FLUOstar OPTIMA, BMG Labtech, Ortenberg, Germany) with approximately 200 nematodes per well in 100 μL, within the visible spectral range of 300–600 nm (firefly luciferase typically emits at 550–570 nm). Luminescence buffer, containing citrate phosphate buffer pH 6.5, 0.1 mM D-luciferin, 1% DMSO, and 0.05% triton-X (all final concentrations), was dispensed automatically into each well [3]. Following the detection of luminescence, the number of nematodes in each well was performed. Subsequently, the mean luciferase activity of nematodes in each well was computed. Three separate experiments with 3–5 replicates total at each timepoint were conducted.

### 4.9. Quantitative Real-Time PCR and Agarose Gel Electrophoresis

L1 stage Caenorhabditis elegans exposed to UVC radiation were either treated with or without 50 µg/mL L-theanine for a duration of 48 h. Total RNA was extracted from approximately 3000 worms using Trizol reagent, followed by cDNA synthesis with oligo dT priming. The expression level of tba-1 was utilized as an internal control to standardize the mRNA levels of the target genes. Samples were analyzed in triplicate, and the primers for the target genes can be found in Appendix A. The final RT-PCR products were visualized through 1.5% agarose gel electrophoresis. Semi-quantitative analysis of the data was conducted using Image J software.

### 4.10. Confocal Microscopy and Image Processing

To capture images of luciferase activity in worms (PE255), they were transferred into 5 μL of luminescence buffer containing 200 μM levamisole for immobilization for a duration of 5–8 min and then mounted on 2% agarose pads in a dark chamber. The specific parameters for photography included ensuring that nematodes (PE255) not treated with luminescent buffer were unable to capture green fluorescence under the imaging luminescence condition. Fluorescence illumination was conducted using FITC filters with an exposure time of 0.2 s. It is important to emphasize that in order to maintain luminescence intensity, nematodes should be imaged within 15 min of exposure to the luminescent buffer.

To analyze changes in the morphology of nematodes’ mitochondria, worms expressing GFP-tagged mitochondria in their body wall muscle (zcIs14 [myo-3::GFP(mit)]) were utilized. The worms were immobilized using levamisole and placed on 2% agarose pads for observation under a Zeiss LSM 710 upright confocal microscope (Carl Zeiss AG, Jena, Germany). Images of mitochondrial morphology were captured at a magnification of 60× using the confocal microscope. Each experiment involved imaging and blind scoring of multiple worms (n ≥ 30). The morphological classifications of mitochondria were established based on findings from a prior study [51]. These classifications included tubular, characterized by a highly interconnected network of elongated/filamentous mitochondria resembling a tube shape; intermediate, consisting of a mixture of interconnected and fragmented mitochondria; and fragmented, denoting small, round mitochondria observed in the captured images. Following this classification, statistical analysis was performed using the chi-square test in IBM SPSS Statistics software version 27.0, and an accumulation histogram was created to visually represent the data.

### 4.11. Statistical Analysis

The results of each experiment underwent verification on a minimum of three occasions. Data were reported as means ± standard deviations, and statistical analyses were conducted using Student’s *t*-test. Significance was defined as *p* < 0.05 or less.

## Figures and Tables

**Figure 1 molecules-29-02691-f001:**
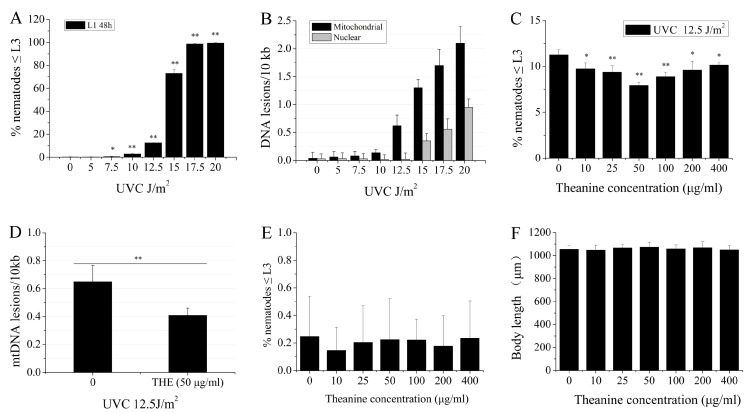
Enhancement of mtDNA damage removal in UVC-exposed *C. elegans* treated with L-theanine. (**A**) The increase in L3 arrest exhibited a dose-dependent relationship after repeated exposure to UVC radiation. (**B**) Following 48 h of UVC exposure, damage to both mitochondrial and nuclear DNA showed a dose-dependent increase after repeated UVC exposure. (**C**) The percentage changes in L3 arrest of UVC-exposed N2 worms treated with varying concentrations of L-theanine are illustrated. (**D**) Treatment with L-theanine for 48 h decreased the frequency of mitochondrial DNA damage induced by UVC exposure. (**E**) The administration of L-theanine did not have an impact on the L3 arrest of N2 worms. (**F**) Treatment with L-theanine did not result in alterations in the body length of N2 worms. Results are means ± SD (results from 3 independent experiments, * *p* < 0.05, ** *p* < 0.01).

**Figure 2 molecules-29-02691-f002:**
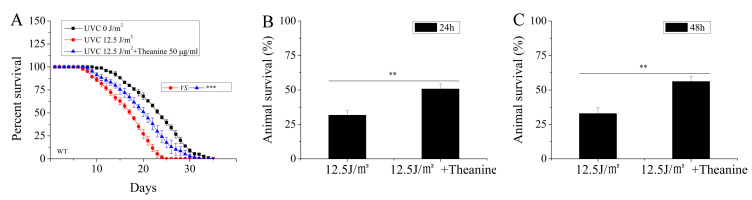
Effect of L-theanine treatment on the lifespan of UVC-exposed *C. elegans*. (**A**) L-theanine treatment significantly extended the lifespan of UVC-exposed nematodes under normal culture conditions. (**B**,**C**) L-theanine treatment for 24 h or 48 h increased the survival rate of nematodes exposed to UVC under heat stress (35 °C for 7 h). Results are means ± SD (3 independent experiments, ** *p* < 0.01, *** *p* < 0.001).

**Figure 3 molecules-29-02691-f003:**
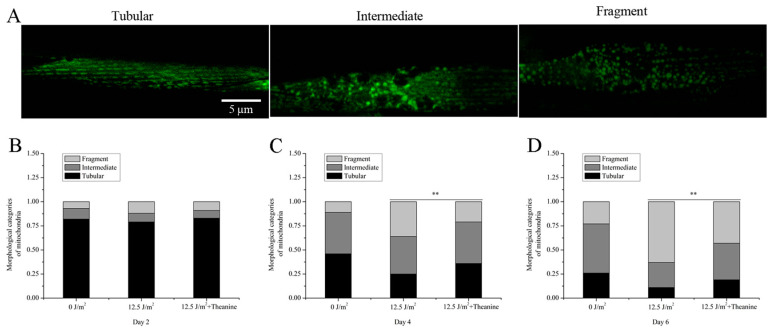
Improvement of mitochondrial morphology in UVC-exposed *C. elegans* treated with L-theanine. (**A**) Representative images of transgenic animals expressing myo-3::matrixGFP in body wall muscle cells. The morphological categorization of mitochondria included tubular, intermediate, and fragmented forms characterized by their respective shapes and connectivity patterns. The assessment of mitochondrial morphology was performed in a blinded fashion to ensure objectivity. (**B**) L-theanine treatment for 2 days did not affect the morphological categories of mitochondria in *C. elegans*. (**C**,**D**) L-theanine treatment for 4 and 6 days improved the morphological categories of mitochondria in UVC-exposed *C. elegans*. Results are means ± SD (at least 3 independent experiments, ** *p* < 0.01).

**Figure 4 molecules-29-02691-f004:**
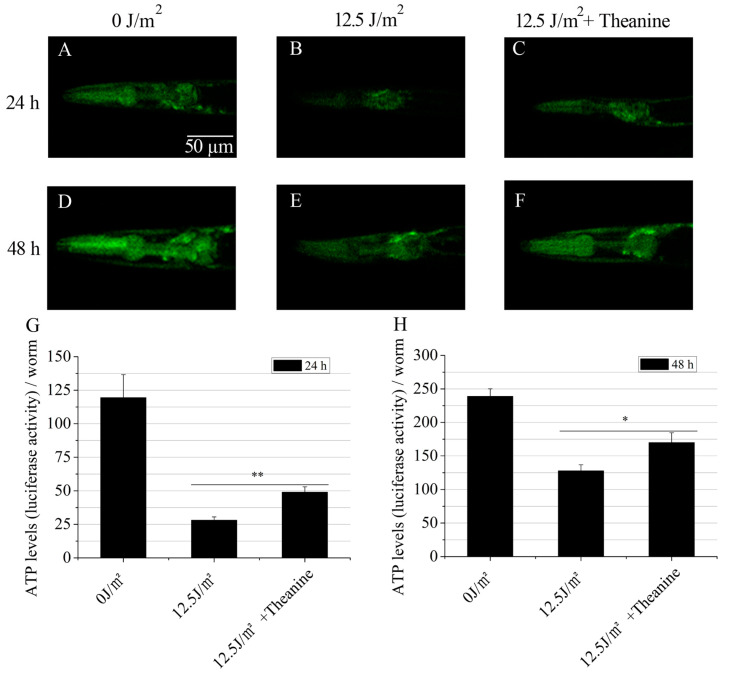
Increasing steady-state ATP levels in UVC-exposed *C. elegans* treated with L-theanine. (**A**–**F**) Representative images of luciferase activity in the head of transgenic animals (PE255) were captured at 24 h and 48 h under complete darkness with 10 s integration after the addition of luminescence buffer and 200 μM levamisole. (**G**,**H**) Steady-state ATP levels were significantly increased in UVC-exposed *C. elegans* treated with L-theanine at 24 h and 48 h compared with UVC treatment. Results are means ± SD (3 independent experiments, *t*-test, * *p* < 0.05, ** *p* < 0.01).

**Figure 5 molecules-29-02691-f005:**
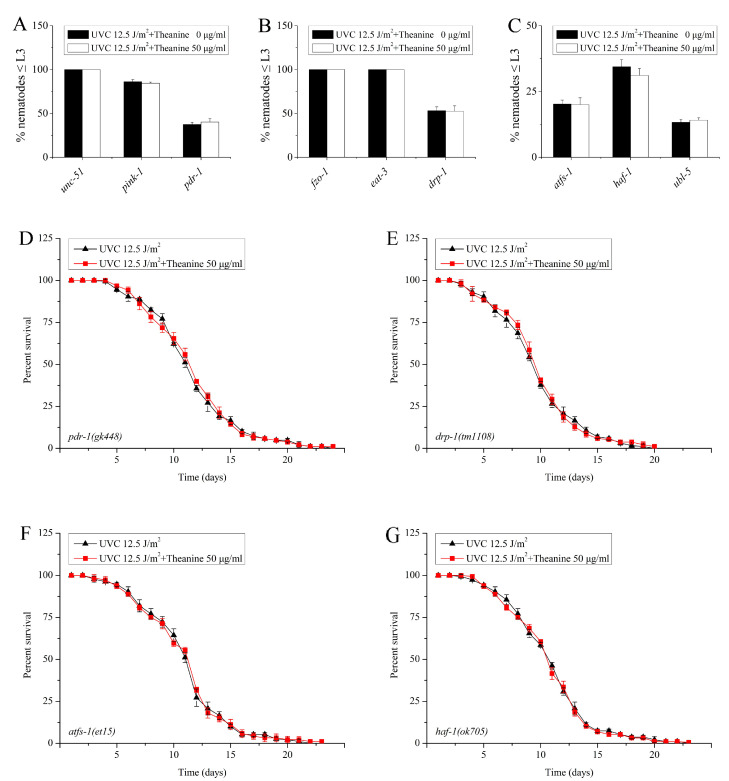
Effect of L-theanine treatment on the L3 arrest and lifespan of UVC-exposed *C. elegans* deleted in autophagy, mitophagy, mitochondrial dynamics, and UPR^mt^ genes. (**A**–**C**) Mutations in autophagy gene (*unc-51*), mitophagy genes (*pink-1* and *pdr-1*), fusion genes (*fzo-1* and *eat-3*), fission gene (*drp-1*), and UPR^mt^ genes (*atfs-1*, *haf-1*, and *ubl-5*) did not affect L3 arrest in UVC-exposed *C. elegans* treated with L-theanine. (**D**–**G**) Mutations in mitophagy genes (*pdr-1*), fission gene (*drp-1*), and UPR^mt^ genes (*atfs-1* and *haf-1*) did not affect the lifespan in UVC-exposed *C. elegans* treated with L-theanine. Results are means ± SD (3 independent experiments, *t*-test).

**Figure 6 molecules-29-02691-f006:**
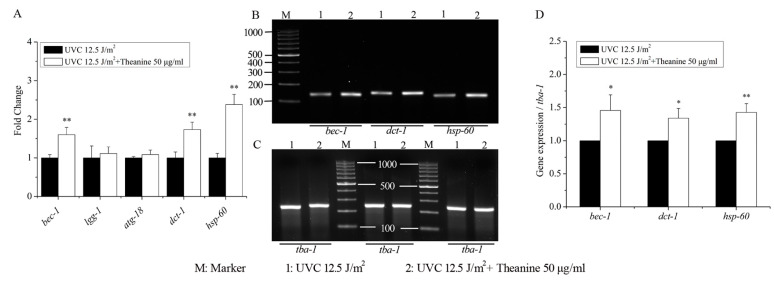
Effect of L-theanine treatment on the relative expression of related genes in UVC-exposed *C. elegans*. (**A**) L-theanine treatment did not alter the mRNA levels of autophagy genes *lgg-1* and *atg-18* in UVC-exposed *C. elegans* treated with L-theanine but upregulated the gene expressions of autophagy (*bec-1)*, mitophagy *(dct-1)*, and UPR^mt^ (*hsp-60*). (**B**,**C**) Agarose gel electrophoresis pictures of *bec-1*, *dct-1*, *hsp-60,* and *tba-1* genes. (**D**) L-theanine treatment significantly increased the expression of *bec-1*, *dct-1,* and *hsp-60* in UVC-exposed *C. elegans*. Results are means ± SD (3 independent experiments, *t*-test, * *p* < 0.05, ** *p* < 0.01).

**Figure 7 molecules-29-02691-f007:**
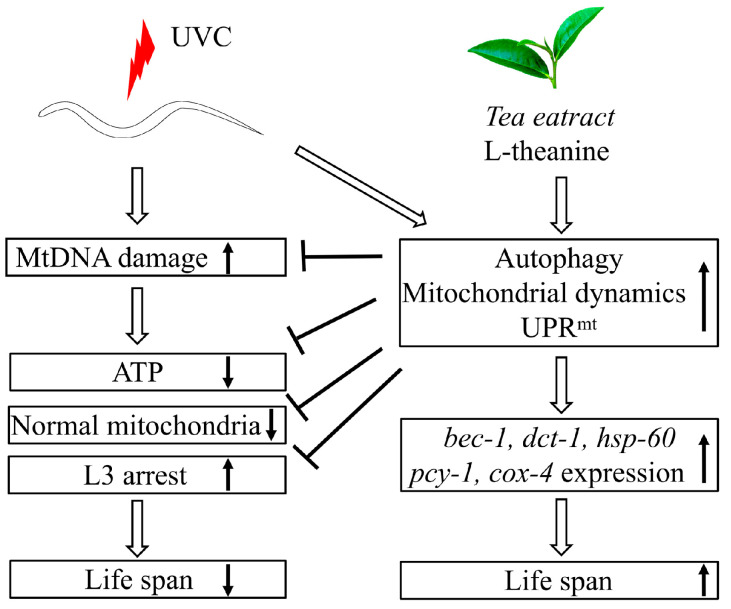
Schematic diagram of the postulated mechanism extending the lifespan of UVC-exposed *C. elegans* treated with L-theanine. Persistent mtDNA damage induced by UVC irradiation causes mitochondrial dysfunction (e.g., reducing ATP levels, increasing larval arrest, and destroying mitochondrial morphology) and shortens the lifespan of *C. elegans*. L-theanine treatment enhances multiple molecular mechanisms (autophagy, mitochondrial dynamics, and UPR^mt^) to remove the mtDNA damage induced by UVC irradiation and improve mitochondrial dysfunction (e.g., increasing ATP levels, reducing larval arrest, and improving mitochondrial morphology) in UVC-exposed *C. elegans*, therefore increasing the lifespan of UVC-exposed *C. elegans*.

## Data Availability

The datasets used and/or analyzed during the current study are available from the corresponding author on reasonable request.

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
