# Peer review of "L-Theanine Prolongs the Lifespan by Activating Multiple Molecular Pathways in Ultraviolet C-Exposed *Caenorhabditis elegans"

_molecules, 2024, doi:10.3390/molecules29112691_

Round 1
Reviewer 1 Report
Comments and Suggestions for Authors
The following are questions about the article:
1. How authors define L1, L2, L3..., L3 larval arrest, L-3 arrest, N2 worms, etc. should be clearly stated in the text.
2. Many abbreviations in the text are not listed in full, e.g., PAH? GFP.
3. C. elegans in the title should be written as Caenorhabditis elegans.
4. Fig.4(G)(H) should use statistical software and method description.
5. L488-489 "quantifying a population of GFP reporter animals, each 20×image was analyzed using Photoshop CS5 " Is it the intensity of the fluorescent light or the size of the shape?
6. The text in Figure 3 is quite blurry and not clear enough to read, so we suggest updating the figure. The authors of Figure 3(A) should point out the location of mitochondrial fragmentation and tubular mitochondria, and what is intermediate? The software and calculation method in Figure 3(B)(C)(D) should be supplemented.
7. The authors should add why [myo-3::GFP(mit)]) was used to assess alterations in mitochondrial morphology.
8. The authors should explain why the transgenic C. elegans (PE255) mutation was used. The effect of sur-5p expression on mitochondrial damage and the specificity of the PE255 nematode mutant species should be added to the article.
9. Figure 6 should be supplemented with electrophoresis grams of the mRNA expression of autophagy (bec-1), mitophagy (dct-1), and UPRmt (hsp-60), as well as the integration software. In addition, the manuscript should add why the authors used these gene expressions for autophagy, mitophagy, mitochondrial dynamics, and UPRmt?
10. It is recommended that Supplementary Figure 3. be included in the manuscript (the results for pcy-1 and cox-4 expression).
Comments on the Quality of English LanguageThis study aimed to prove that L-theanine treatment increased ATP production and improved mitochondrial morphology to extend the lifespan of UVC-exposed nematodes. The experimental results obtained are good, however, the correlation between the upstream and downstream mechanisms of many related genes is not clearly explained. In addition, the authors should explain how genes such as nc-51, pink-1, pdr-1, fzo-1, eat-3, and drp-1 affect autophagy, mitophagy, fusion, and fission.
Author Response
Thanks for your comments, the responses have been attached.

Reviewer 2 Report
Comments and Suggestions for Authors
The authors showed that in C. elegans exposed to UVC, theanine activated several pathways such as autophagy, mitophagy, and mitochondrial unfolded protein response (UPRmt), and prolonged life span.
These results are clear and very interesting, but this paper needs minor revision.
In their discussion, the authors described the action of antioxidants on mitochondrial DNA and referenced green tea extract [Ref. 43]. However, the antioxidant effect of green tea extract is thought to be mainly due to catechins, and it is unlikely that theanine has a strong antioxidant effect.
The last sentence of the discussion needs to be deleted or rewritten.
Theanine is indeed an important component of green tea, but green tea also contains caffeine and catechins, which can have antagonistic effects against theanine. In addition, not all teas contain theanine. Therefore, it is not appropriate to express that the effects of theanine are expected from tea drinking.
Author Response

(The authors gave the same response as above.)
